# China's Eco-Efficiency: Regional Differences and Influencing Factors Based on a Spatial Panel Data Approach

**Liangen Zeng** 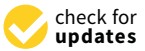

College of Urban and Environmental Sciences, Peking University, Beijing 100871, China; zengliangen@pku.edu.cn; Tel.: +10-62768258

**Abstract:** From the Kyoto Protocol to the Copenhagen Conference and the Paris Agreement, eco-environmental problems have gradually become a matter of common concern worldwide. Eco-efficiency (EE) is an essential indicator for measuring levels of sustainable development. This study uses an epsilon-based measure (EBM) model with undesirable outputs to evaluate the EEs of 30 Chinese provinces during the research period 2008 to 2017, and a spatial Durbin model (SDM) to search for the impact factors of EE. The results indicate that most provinces in China have a low EE level. The EE value of the eastern area is higher than are those for the central, western, or northeastern areas. The EE in China as a whole demonstrates an inverted V-shaped trend with a high point in 2011. The SDM shows that economic development level, foreign trade dependence, and technological progress exert significant positive effects on EE, while population density exerts significant negative influences on EE. This paper provides scientific bases for the formulation of policies resulting in sustainable development.

**Keywords:** Eco-efficiency; the EBM model with undesirable outputs; spatial Durbin model

---



## 1. Introduction

The global mean temperature has risen 0.85 °C between 1880 and 2012, and this was mainly caused by human activities [1]. According to the IPCC report, average temperatures across the world are likely to rise by 4 °C above pre-industrial levels without effective action to inhibit greenhouse gas emissions [2]. In this context, the Paris Agreement was signed to prevent the global mean temperature from rising to more than 2 °C above pre-industrial levels, while working to curb the rise to 1.5 °C above pre-industrial levels [3]. Therefore, probing the relationship between economic growth and the resource environment is of great significance for sustainable global development.

Eco-efficiency (EE) is an essential indicator for measuring levels of sustainable development [4]; it was first proposed by Schaltegger and Sturm [5] and emphasized the creation of more goods and services while consuming fewer resources and producing less waste and pollution [6]. This accords with the core idea of sustainable development, which fosters the harmonious development of the economy, resources, and environment.

In recent decades, China has undergone rapid and stable economic growth. China already has, in fact, the second-largest economy in the world, and its share of global GDP passed the 14.8% mark in 2018 [7]. However, the economic growth of China still largely bears the cost of sacrificing the ecological environment [8]. China is the largest consumer of energy, accounting for 23.2% of global energy consumption in 2018 [9], and also is the most significant source of $CO_2$ emissions, producing 21.2% of global $CO_2$ emissions in 2016 [10]. According to the 2018 Environment Performance Index (EPI), China's EPI ranked 120 in all the 180 countries and regions in 2018 [11].

Facing the ever-increasingly severe problems of both global warming and the ecological environment, the Chinese government has proposed a goal of building a beautiful China and setting up a five-sphere integrated plan; this requires that, while taking economic construction as the central task, there must also be comprehensive advances of

the construction of political, economic, cultural, social and ecological development [12]. However, China also faces the problem of vast territories with regional differences regarding resource endowment, technology accumulation, and capacity for environmental governance. Therefore, the formulation of a policy for pollutant reduction in different regions must not apply the "one size fits all" approach. In 2011, the Chinese government promulgated the main functional area planning, which has required the implementation of different environmental strategies in different regions [13]. The environmental policies should be designed separately to account for the concrete EE conditions in various areas. What then are the characteristics of regional EE? What are the primary factors that lead to changes in EE? Only by recognizing and mastering these challenges can policymakers avoid establishing ineffective or harmful strategies for eco-environmental improvement.

In general, methods for EE evaluation can be divided into the stochastic frontier analysis (SFA) and data envelopment analysis (DEA) methods. However, traditional DEA methods, such as the Charnes–Cooper–Rhodes (CCR) model, Banker–Charnes–Cooper (BCC) model, and the slacks-based measure (SBM) model, have their own limitations. CCR and BCC neglect the non-radial slacks; this limitation may cause a biased measure of the efficiency score of the decision-making unit (DMU). The SBM model requires input or output variables to change proportionally, and cannot cope with other cases properly [8,14–22]. The epsilon-based measure (EBM) model with undesirable outputs is a recent approach to evaluate the EE, which can compensate for the weakness of traditional DEA methods as well as deal with undesirable outputs [15–26]. Additionally, the common methods for analyzing the factors driving EE, the Tobit model, Spatial Lag Model (SLM), and Spatial Error Model, also have their own strengths and weaknesses. The traditional Tobit model fails to consider spatial factors [23]. Although spatial effects are involved in the Spatial Lag Model (SLM) and Spatial Error Model, the lags in both spatial independent variables and the dependent spatial variable are not considered at the same time in these models [27–29].

The major contributions of this article include: (1) The factors influencing EE were determined by using the Spatial Durbin model (SDM), which overcomes the drawbacks of both SEM and SLM, and includes spatial lags in the explained variable and independent variables [27,30]. (2) This study uses the empirical research results to propose a series of development policies designed to accomplish the targets of both the energy savings and emissions reductions, and suggestions for future research are also provided herein.

The rest of the article is as follows: In Section 2, the relevant researches are reviewed and evaluated. In Section 3, the article introduces the EBM model with undesirable outputs and the SDM method. In Section 4, indicator selection and data sources are described. In Section 5, the paper analyzes the characteristics of Chinese EE levels and explores their causes. Section 6 involves concluding the research results and offering policy suggestions. The flowchart is given in Figure 1.

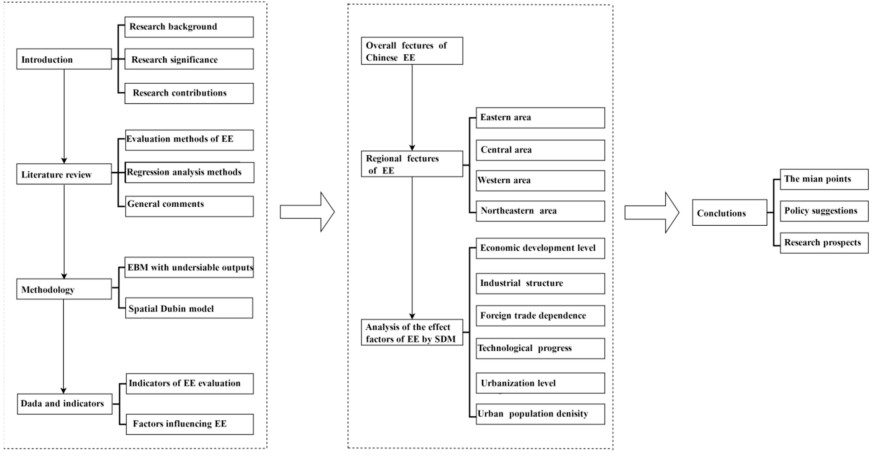

**Figure 1.** Flowchart of the empirical research of Eco-efficiency (EE) in China.

## 2. Literature Review

EE evaluation methods can, in general, be divided into the stochastic frontier analysis (SFA) and data envelopment analysis (DEA) approaches. SFA estimates a parametric frontier of the best possible practices given a standard cost or profit function [31] and is applied in EE measurement of some economic sectors [32–34]. However, as a parametric method, it requires the evaluated parameters to be independent, which is very difficult to keep consistent with reality [31,35–38].

The DEA method was established by Charnes et al. [39] and is a model for calculating the efficiency score of a DMU; it does not require any prior assumption about the production frontier and can deal with complex systems that have multiple inputs and outputs [23]. Hence, DEA is widely utilized in studying EE (Table 1). The traditional DEA models can be divided into two categories: radial models, such as the CCR and BCC models, and non-radial models. For instance, Fan et al. [40], Pai et al. [41], Moutinho et al. [42], Rybaczewska-Błaże and Gierulski [43], and Shah and Dong [44] have applied the radial models to calculate the EE at different spatial levels. However, such radial DEA models fail to take the effect of nonradial slacks on the technical efficiency into account and cannot realize the factor decomposition when calculating the technical efficiency, which can cause biased estimation results [8]. To capture the slack factors, a non-radial DEA model called the slack-based measure (SBM) was proposed by Tone [45]; however, it does not consider undesirable outputs. In 2003, Tone proposed the SBM with an undesirable outputs model that can capture slack factors and also incorporate pollutant emissions as the undesirable output factor [46]. Therefore, the SBM with undesirable outputs model is the most common in calculating EE across sectors and industries [47–49]. However, in the SBM methods, the slacks are not necessarily proportional to the inputs or outputs, the DMU may lose the proportionality in the original inputs or outputs [50]. Based on the shortcomings of the SBM methods, Tone and Tsutsui [17] proposed the epsilon-based measure (EBM), which can take radial and non-radial factors into account simultaneously. Yang et al. [16] applied the EBM DEA model to calculate the ecological energy efficiency of in the Chinese 30 provinces from 2007 to 2015, but the pollutants, such as $SO_2$ and NOx, were used as input indicators. To estimate the Urban EE in China, Chen et al. [51] proposed the EBM model with undesirable outputs, which can combine both radial and non-radial features into a composite model and also deal with undesirable factors.

**Table 1.** Summary of data envelopment analysis (DEA) applications to the assessment of eco-efficiency.

| Author | Methodology | Objects and Period | Variables |
|---|---|---|---|
| Fan et al. [40] | CCR and BCC DEA models | The eco-efficiency levels of 40 Chinese industrial parks in 2012 | Input: Land, Energy, Water Desirable output: Industrial value added Undesirable output: Wastewater, Solid waste, COD, $SO_2$ |
| Pai et al. [41] | CCR and BCC DEA models | Th eco-efficiencies of 60 industrial parks in Taiwan | Input: Site area, Labor force, Electricity, Water, Waste discharge, Airborne, Particles Output: The overall operating income |
| Moutinho et al. [42] | BCC DEA model | The eco-efficiency of the 16 Latin American countries from 1994 to 2013 | Input: Population density, labor productivity, Energy, Renewable energy, Gross capital formation productivity Output: The inverse ratio of carbon intensity |
| Rybaczewska-Bła˙zejowsk and Gierulski [43] | Life cycle assessment (LCA) and BCC DEA model | Th eco-efficiencies of agricultural production in 28 member states of the European Union in 2015 | Input: Labor, Capital, Energy Desirable output: GDP Undesirable output: $SO_2$ |
| Shah et al. [44] | CCR and BCC DEA models | The eco-efficiency at the industrial park/complex level of Ulsan metropolis and Korea in 2005, and 2010, and 2015 | Input: Land, Labor force, Energy Output: Gross output |

**Table 1.** *Cont.*

| Author | Methodology | Objects and Period | Variables |
|---|---|---|---|
| Peng et al. [47] | The SBM DEA model with undesirable outputs | The eco-efficiency of the Huangshan National Park in China from 1981 to 2014 | Input: Average wage level of employees, New fixed asset investment, Energy, Water, Desirable output: Per capita tourism income Undesirable output: Garbage, Sewage, Waste gas |
| Ning et al. [48] | The SBM DEA model with undesirable outputs | The eco-efficiency of state-owned forestry enterprises in Northeast China from 2003 to 2015 | Input: Labor, Capital, Land Desirable output: Total output, Sale Undesirable output: Effluent, Exhaust, Solid-waste |
| Zheng et al. [49] | The SBM DEA model with undesirable outputs | The eco-efficiency of the Chinese 31 provinces from 2000 to 2015 | Input: Water footprint; Labor force; Capital, Cost of resource and environment, Land Desirable output: GDP Undesirable output: Gray water footprint, Environmental pollutants |
| Wang et al. [50] | The SBM DEA model with undesirable outputs | The eco-efficiency of regional tourism in Chinese 31 provinces from 1997 to 2016 | Input: Labor, Capital, Water, Energy Desirable output: Revenue from tourism Undesirable output: Tourism effluent discharge Tourism waste discharge Tourism $SO_2$, Tourism $CO_2$, |
| Yang et al. [16] | The EBM DEA model | The ecological energy efficiency of in Chinese 30 provinces from 2007 to 2015 | Input: Labor, Capital, Energy, $SO_2$, $NO_X$ Desirable output: GDP |
| Chen et al. [51] | The EBM DEA model with undesirable outputs | The ecological efficiency of in Chinese 259 cities from 2007 to 2016 | Input: Labor, Capital, Energy, Water, Land Desirable output: GDP Undesirable output: Industrial discharged wastewater, Industrial sulfur dioxide emission, Industrial soot (dust) emission |

The mainstream methods for exploring the impact of EE are divided into two categories: one is the Tobit model, which is represented by a non-spatial panel data model, and the other is the conventional spatial panel data model, which is represented by the Spatial Lag Model (SLM) and Spatial Error Model (SEM). The Tobit model is applicable when the dependent variable may be truncated [52]. Wang et al. [53], Ma et al. [54], Zhu et al. [55], Liu et al. [56], Zhong et al. [57], and Dong et al. [58] have applied the Tobit method to analyze related EE factors. However, according to the theory of the First Law of Geography, everything is related to everything else, but near things are more related than distant things [59]. Therefore, the non-spatial Tobit model omits the spatial correlation, and the estimation results may exhibit some deviation from real results when there are spatial correlations among variables [23,59]. The conventional spatial panel data models, such as SLM or SEM, can consider the spatial factors. SLM is also known as a spatial autoregressive model [60]. SLM contains a spatially lagged explained variable and hypothesizes that the explained variable in the local area can be influenced by the explained variable in the neighboring areas. SEM includes a lagged error term and hypothesizes that the errors in neighboring areas are likely to interact [61]. The two spatial methods have been widely employed in analyzing the causal factors of EE in different countries and regions [62–67].

This literature review shows that SLM or SEM have been the main methods used to explore the factors influencing EE. However, SLM only contains the lag term of the spatial explained variable, and SEM only introduces error terms into spatial autocorrelation [68]. Therefore, they cannot take the spatial interaction of explanatory variables into account.

The breakthrough of the research approach is as follows: Spatial Durbin model (SDM) is applied to investigate the factors affecting EE. This was proposed by Lesage and Pace [27]. SDM contains both the spatially lagged dependent variable and independent variables, not only taking the spatial dependence of the explained variable into account but also that of the explanatory variables [27,68].

## 3. Methodology

### 3.1. The EBM DEA Model with Undesirable Outputs

The DEA is a non-parametric analysis approach used to calculate the technical efficiency of DMUs. The traditional DEA models are generally divided into two categories: Radial models, such as CCR and BCC models, or Non-radial models, such as an SBM model. Radial models ignore the slack variables and require that the inputs change in the same proportion, which often does not conform with reality. Non-radial SBM models, which capture the non-radial slacks, seek to maximize the input and output inefficiencies by identifying the points farthest from the frontier [69]. However, they may lose the original proportion of the input because of the slack variable. To resolve these problems, Tone and Tsutsui [17] proposed the EBM (Epsilon-Based Measure) DEA model, which can combine the advantages of both radial and non-radial DEA methods. The standard EBM model can exhibit the proportionality between the target value and the actual value and adequately reflect the contrast between the non-radial part of inputs or outputs [70]. This gives full consideration to the undesirable output factors that are by-products accompanying the economic output. Based on Chen et al. [51], and Wu et al. [69], and Ren et al. [71], the EBM DEA model with undesirable outputs is expressed as

$$\kappa^* = \min \left[ \frac{\alpha - \varepsilon_x \sum_{i=1}^{m} \frac{\omega_i^- s_i^-}{x_{i0}}}{\beta + \varepsilon_y \sum_{r=1}^{S} \frac{\omega_r^{+g} s_r^{+g}}{y_{r0}} + \varepsilon_b \sum_{p=1}^{q} \frac{\omega_p^{-b} s_p^{-b}}{b_{p0}}} \right] s.t. \begin{cases} \sum_{j=1}^{n} x_{ij}\lambda_j + s_i^{-b} = \alpha x_{i0} & i = 1,2,\ldots,m \\ \sum_{j=1}^{n} y_{rj}\lambda_j - s_r^{+g} = \beta y_{r0} & r = 1,2,\ldots,s \\ \sum_{j=1}^{q} b_{pj}\lambda_j + s_p^{-b} = \beta b_{p0} & p = 1,2,\ldots,p \\ \lambda_j \geq 0, s_i^{-b} \geq 0, s_r^{+g} \geq 0, s_p^{-b} \geq 0 \end{cases} \tag{1}$$

Here $\kappa^*$ is the efficiency score, and it ranges from [0, 1]; Variables $m$, $s$, and $q$ stand for the numbers of inputs, desirable outputs, and undesirable outputs, respectively. Parameters $x_{i0}$, $y_{r0}$, and $b_{po}$ represent input $i$, desirable output $r$, and undesirable output $k$ of the decision-making unit$_0$ (DMU$_0$), respectively. $s_i^-$, $s_r^{+g}$ and $s_p^{-b}$ indicate the slacks of the $i$th input, the $r$th desired output, and the $p$th undesired output, respectively. $\omega_i^-$ is $i$th input weight, $\sum \omega_i^- = 1 (\forall i \omega_i^- \geq 0)$. $\omega_r^{+g}$ and $\omega_p^{-b}$ represent the desired output weight and the undesired output weight, respectively. $\varepsilon_y$ denotes the set of radial $\beta$ and non-radial slacks; $\varepsilon_x$ and $\varepsilon_y$ meet the conditions: $0 \leq \varepsilon_x \leq 1$ and $0 \leq \varepsilon_y \leq 1$.

### 3.2. Global Moran's I

The existence of spatial autocorrelation for EE can be verified by Global Moran's I and its formula is measured as follows:

$$Global\ Moran'\ I = \frac{\sum_{i=1}^{N} \sum_{j=1}^{N} W_{i,j} \left( EE_{i,t} - \overline{EE_t} \right) \left( EE_{j,t} - \overline{EE_t} \right)}{\left[ \frac{1}{N} \sum_{i=1}^{N} \left( EE_{i,t} - \overline{EE_t} \right)^2 \right] \sum_{i=1}^{N} \sum_{j=1}^{N} W_{i,j}} \tag{2}$$

where $W$ indicates spatial matrix; in this paper, if province $i$ and $j$ are adjacent, $W_{i,j} = 1$ and not adjacent to 0. $N$ represents the total amount of research provinces. $\overline{EE}$ represents the mean value of the EE. The range of Global Moran's I is $[-1, 1]$. When Global Moran's I > 0 means that EE has a positive spatial correlation. If Global Moran's I < 0, it indicates that the stronger the negative correlation in the spatial distribution. When Global Moran's I = 0, the EE of different provinces demonstrates spatially as independent or random distributions.

The significance of Global Moran's I can be examined by Z-test, which can be calculated as follows:

$$Z = \frac{I - E(I)}{\sqrt{V - (I)}} \tag{3}$$

where *E(I)* and *V(I)* stand for the expected value and variance of Global Moran's I, respectively. If $|Z| > 1.96$ or $|Z| > 2.54$, indicates spatial autocorrelation that is significant at the 0.05 or 0.01 confidence level, respectively.

### 3.3. Spatial Durbin Model

There are marked spatial correlations among adjacent geographical units with respect to certain phenomena of economic geography or certain specific property values [59]. The traditional spatial regression equation is constructed as follows:

$$\begin{cases} Y = \rho WY + \beta X + u \\ \quad u = \lambda Wu + \varepsilon \\ \quad \varepsilon \sim N(0, \ \sigma^2 In) \end{cases} \tag{4}$$

Here, $\rho WY$ refers to the spatial lag of the explained variable, $\rho$ represents the spatial autoregression coefficient, $WY$ denotes the spatial lag explained variable, $\lambda Wu$ is the spatial error term, $\lambda$ indicates the autoregressive parameter, and $u$ and $\varepsilon$ are both error perturbations [30].

When $\rho = 0$ and $\lambda = 0$, model (4) is a conventional linear model, suggesting that the explained variable has no relationship to the spatial effects. If $\rho = 0$ and $\lambda \neq 0$, model (4) becomes the SEM, implying that an explained variable is randomly affected by adjacent areas. When $\rho \neq 0$ and $\lambda = 0$, model (4) becomes the SLM. In this model, the spatial autocorrelation can be explained by a spatially lagged dependent variable in the SLM.

However, there are obvious shortcomings in both SLM and SEM. The dependent variable may be explained, not just through a lag effect of the dependent variable or spatially autocorrelated error term, but also through a spatially lagged explained variable and spatially lagged explanatory variables simultaneously. The SDM combines the characters of SLM and SEM, can include the spatial interaction effects from both dependent and independent variables [72], can also inspect the effect of the variable error on observation values, and can then yield a more accurate result [27]. Therefore, this study applies the SDM to analyze the influential factors of EE, which is presented in Equation (5):

$$Y = \rho WY + \beta X + \theta WX + \varepsilon \tag{5}$$

In Equation (5), $W$ represents spatial matrix; $\theta WX$ stands for the spatial lag of the spatial lagged independent variables [30]; the definitions of the other variables in Equation (5) are consistent with those in Equation (4).

## 4. Data Source and Indicator Selection

This study consists of evaluations of EE and influencing factors in China, using panel data for 30 continental provinces from 2007 to 2018. Data from Tibet were not included. As shown in Figure 2, the paper divides these provinces into four regions in accordance with China's National Bureau of Statistics.

### 4.1. Indicators of EE Evaluation

Following Fan et al. [40], Zhang et al. [73], Yang and Zhang [74], and Yu et al. [75], the study chooses capital stock, the total number of employees, total water consumption, and total energy consumption as input indicators. Chinese agencies have not published data on provincial capital stock, and these data must, therefore, be calculated. To do so, the paper applied the perpetual inventory method as follows: $Ci,t = Ii,t + (1 − δ)Ci,t − 1$. C and I represent the capital stock and gross fixed capital formation, respectively. The subscripts $i$ and $t$ indicate the province $i$ and year $t$, respectively, and $δ$ is the depreciation rate of capital stock. Consistent with Zhang [76], the paper suggested the value of $δ$ is 9.6% and calculated capital stock in 2010 as equal to the value of the gross fixed capital formation in 2004 divided by 10%. The capital stock data were converted into 2008 constant prices. The data on gross fixed capital formation, the total number of employees, and total water

consumption, and urban construction land were taken from the China Statistical Yearbook (CSY) (2009–2018) [76]. The data on energy consumption come from the China Energy Statistical Yearbook (2009–2018) [77].

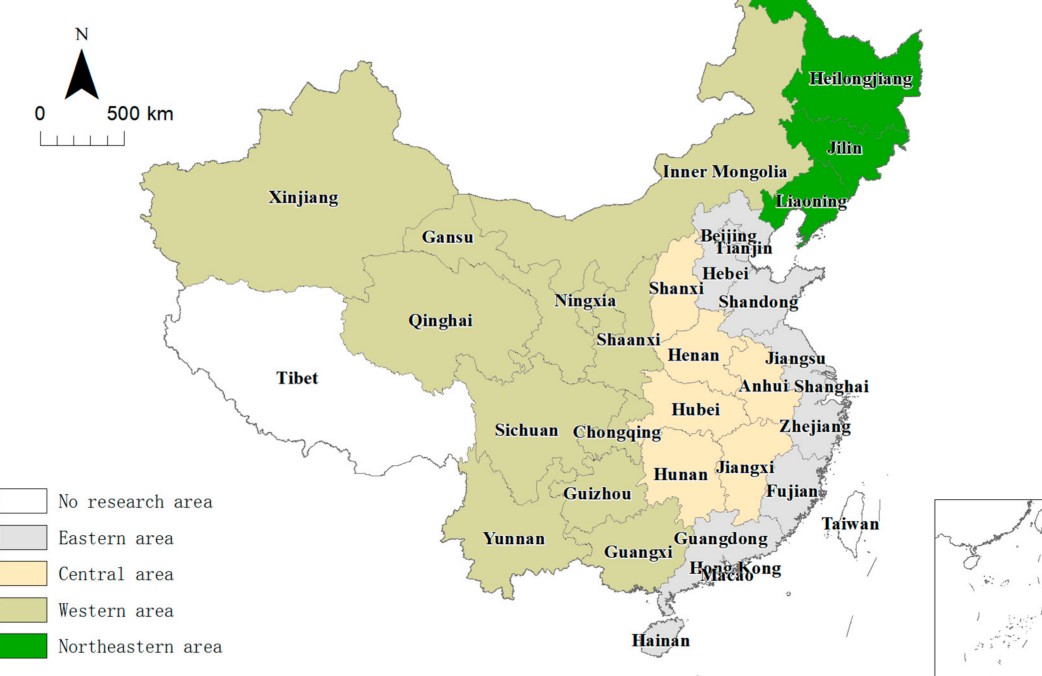

**Figure 2.** The schematic diagram of four economic zones of China.

The article selected GDP as the desired output. According to the total major pollutant emission reduction assessment methods published by the State Council of China in 2007, $SO_2$, smoke and dust, COD (Chemical Oxygen Demand), and ammonia nitrogen are the key monitoring pollutants; $CO_2$, a major cause of greenhouse gas, is also a cause of climate change and global warming. Therefore, the paper identified them as undesirable outputs. The data on $CO_2$ were collected from China Emission Accounts and Datasets (2020) [78]. The other data on pollution emission were collected from the CSY (2009–2018) [76]. Table 2 shows the summary statistics of the variables.

**Table 2.** EE measurement index system.

| Primary Indicators | Secondary Indicators | Mean | Std. Dev | Min | Max |
|---|---|---|---|---|---|
| Inputs | Capital stock (unit: $10^8$ yuan) | 68,357.9 | 46,755.2 | 5832 | 231,280 |
| | Labor force (unit: $10^4$) | 2666.7 | 1744.7 | 301 | 6767 |
| | Total energy consumption (unit: $10^4$ tons of SCE) | 13,624.5 | 8157.9 | 1135 | 38,899 |
| | Total water consumption (unit: $10^8$ L) | 201 | 142 | 22.3 | 591.3 |
| | Urban construction land (unit: sq.km) | 1567.1 | 1078.1 | 109 | 5577 |
| Desired outcomes | GDP (unit: $10^8$ yuan) | 18,559.3 | 15,297.9 | 1019 | 80,956 |
| Undesired outcomes | $CO_2$ emissions (unit: $10^6$ tons) | 300.5 | 192.2 | 25 | 842 |
| | $SO_2$ emissions (unit: $10^4$ tons) | 63 | 40.7 | 1.43 | 182.7 |
| | Smoke and dust emissions (unit: $10^4$ tons) | 58.9 | 43. | 5.75 | 198.3 |
| | COD emissions (unit: $10^4$ tons) | 43.4 | 31.1 | 1.47 | 179.8 |
| | Ammonia nitrogen (unit: $10^4$ tons) | 6.25 | 4.51 | 0.56 | 23.09 |

### 4.2. Factors Influencing EE

The higher the regional economic development level, the greater is the capacity for environmental governance. Consulting Zhu et al. [55], Dong et al. [58] Xu [79], and

Liu et al. [80], the article selected GDP per capita to represent the level of economic development. The data on GDP per capita were converted into the 2008 constant price. The anticipation is positive.

The index of the industrial structure reflects the distribution of industry in an area. The higher the level of industrial structure, the more effective is the allocation of resources. Rational allocation of resources, to an extent, promotes stable and healthy development in the regional ecological environment. Therefore, with reference to Zhu et al. [55], Zhou et al. [81], and Shi and Wang [82], the study has selected the ratio of the tertiary industry to provincial GDP as the index for measuring industrial structure and expect that this has positive effects on EE.

Developing foreign trade can effectively encourage mechanisms of competition, promote rational allocation of resources, and improve technical progress in local enterprises. Following Ma et al. [54], Xu [79], Liu et al. [80], and Zhou et al. [83], the study selected the ratio of the total foreign trade volume to the provincial GDP to evaluate dependence on foreign trade.

Advanced production technology can improve the efficiency of resource utilization, optimize industrial infrastructure, and improve EE. The scale and intensity indicators of research and development (R&D) activities are generally used to measure the strength and core competitiveness of science and technology. Based on the work of Dong et al. [58], Liu et al. [80], and Zhou et al. [84], the study selected the ratio of R&D expenditure to provincial GDP for measuring the level of technological progress.

As a consequence of urbanization, the demand for energy and resources increases, resulting in more emissions of pollutants. This, in turn, becomes a barrier to improving EE [85,86]. Therefore, the influence of urbanization on EE should be negative.

Population density exerts important impacts on the regional economy and environmental protection. An increase in population density increases the labor force and the number of consumers, which can promote economic growth; however, it also results in greater consumption of resources and increases the pressure on the ecological environment. Following Guan and Xu, [62], Zhou et al. [80], Díaz-Villavicencio et al. [87], and Chen et al. [88], the study selected population density as a control variable (Table 3).

**Table 3.** Influencing factors.

| Explanatory Variable | Variables' Definition and Unit | Key References | Pre-Judgment |
|---|---|---|---|
| Economic development level (EDL) | GDP per capita ($10^4$ RMB) | [55,58,79,80] | Positive |
| Industrial structure (IS) | The proportion of the added value of the tertiary industry to provincial GDP (%) | [55,81,82] | Positive |
| Foreign trade dependence(FTD) | The proportion of the total import and export trade to provincial GDP (%) | [54,79,80,83] | Unknown |
| Technological progress (TP) | Proportion of R&D expenditure to provincial GDP (%) | [58,80,84] | Positive |
| Urbanization level (UL) | The proportion of city population in total population (%) | [85,86] | Negative |
| Population density(PD) | Ratio of the total regional population to regional area (person/sq.km) | [62,84,87,88] | Unknown |

## 5. Empirical Analysis

### 5.1. Overall Characteristics of Chinese EE

The EE of each province from 2008 to 2017 was calculated using the EBM model with undesirable outputs, and results are shown in Table 4.

Table 4 shows the indices for the EEs of the 30 provinces of China for the period 2008 to 2017. From the table, we can see that the overall development level of EE in China is relatively low that the average EE of the 30 provinces in China is 0.674, which does not reach the efficiency frontier, and there is considerable room for development. In comparing regions, one observes that the Eastern area is the most highly developed region, the Central

area is the second most highly developed, and it is followed by the Western area, and the Northeast area is the lowest; (Figure 3).

**Table 4.** EE of 30 provinces in China from 2008 to 2017.

| Regions | Provinces | 2008 | 2009 | 2010 | 2011 | 2012 | 2013 | 2014 | 2015 | 2016 | 2017 | Mean |
|---|---|---|---|---|---|---|---|---|---|---|---|---|
| | Beijing | 1 | 1 | 1 | 1 | 1 | 1 | 1 | 1 | 1 | 1 | 1 |
| | Tianjing | 1 | 1 | 1 | 1 | 1 | 1 | 1 | 1 | 1 | 1 | 1 |
| | Hebei | 0.751 | 0.696 | 0.732 | 0.738 | 0.69 | 0.666 | 0.634 | 0.643 | 0.641 | 0.593 | 0.678 |
| | Shanghai | 1 | 1 | 1 | 1 | 1 | 1 | 1 | 1 | 1 | 1 | 1 |
| East | Jiangsu | 0.889 | 0.884 | 0.893 | 0.855 | 0.811 | 0.789 | 0.78 | 0.761 | 0.773 | 0.75 | 0.818 |
| | Zhejiang | 1 | 1 | 1 | 0.905 | 0.874 | 0.814 | 0.805 | 0.861 | 0.893 | 0.831 | 0.898 |
| | Fujian | 1 | 1 | 1 | 0.852 | 0.792 | 0.771 | 0.755 | 0.743 | 0.778 | 0.7 | 0.839 |
| | Shandong | 0.772 | 0.759 | 0.8 | 0.783 | 0.734 | 0.738 | 0.697 | 0.695 | 0.713 | 0.682 | 0.737 |
| | Guangdong | 1 | 1 | 1 | 1 | 1 | 1 | 1 | 1 | 1 | 1 | 1 |
| | Hainan | 0.569 | 0.645 | 0.607 | 0.635 | 0.631 | 0.581 | 0.589 | 0.509 | 0.558 | 0.56 | 0.588 |
| | **Mean** | **0.898** | **0.898** | **0.903** | **0.877** | **0.853** | **0.836** | **0.826** | **0.821** | **0.836** | **0.812** | **0.898** |
| | Shanxi | 0.66 | 0.619 | 0.633 | 0.662 | 0.614 | 0.597 | 0.57 | 0.564 | 0.546 | 0.517 | 0.598 |
| | Anhui | 0.535 | 0.543 | 0.591 | 0.62 | 0.605 | 0.597 | 0.569 | 0.582 | 0.591 | 0.558 | 0.579 |
| Central | Jiangxi | 0.623 | 0.624 | 0.659 | 0.684 | 0.649 | 0.632 | 0.622 | 0.612 | 0.619 | 0.581 | 0.631 |
| | Henan | 0.703 | 0.686 | 0.706 | 0.708 | 0.662 | 0.644 | 0.614 | 0.622 | 0.652 | 0.626 | 0.662 |
| | Hubei | 0.565 | 0.59 | 0.599 | 0.635 | 0.613 | 0.625 | 0.574 | 0.618 | 0.626 | 0.563 | 0.601 |
| | Hunan | 0.657 | 0.625 | 0.69 | 0.736 | 0.713 | 0.706 | 0.68 | 0.715 | 1 | 0.661 | 0.718 |
| | **Mean** | **0.624** | **0.614** | **0.646** | **0.674** | **0.643** | **0.633** | **0.605** | **0.619** | **0.672** | **0.584** | **0.624** |
| | Inner Mongolia | 0.666 | 0.665 | 0.661 | 0.674 | 0.639 | 0.632 | 0.604 | 0.637 | 0.699 | 0.658 | 0.653 |
| | Guangxi | 0.55 | 0.607 | 0.639 | 0.664 | 0.601 | 0.582 | 0.567 | 0.566 | 0.565 | 0.525 | 0.587 |
| | Chongqing | 0.629 | 0.616 | 0.649 | 0.68 | 0.702 | 0.705 | 0.672 | 0.666 | 0.678 | 0.647 | 0.664 |
| | Sichuan | 0.655 | 0.637 | 0.672 | 0.703 | 0.672 | 0.661 | 0.63 | 0.627 | 0.616 | 0.58 | 0.645 |
| | Guizhou | 0.551 | 0.549 | 0.619 | 0.622 | 0.588 | 0.573 | 0.554 | 0.544 | 0.527 | 0.469 | 0.56 |
| West | Yunnan | 0.598 | 0.598 | 0.623 | 0.617 | 0.6 | 0.604 | 0.564 | 0.555 | 0.538 | 0.491 | 0.579 |
| | Shaanxi | 0.683 | 0.782 | 1 | 1 | 0.718 | 0.682 | 0.654 | 0.664 | 0.682 | 0.621 | 0.749 |
| | Gansu | 0.508 | 0.51 | 0.546 | 0.557 | 0.541 | 0.542 | 0.512 | 0.523 | 0.524 | 0.507 | 0.527 |
| | Qinghai | 0.604 | 0.6 | 0.633 | 0.654 | 0.599 | 0.538 | 0.499 | 0.49 | 0.502 | 0.475 | 0.559 |
| | Ningxia | 0.364 | 0.368 | 0.413 | 0.413 | 0.397 | 0.391 | 0.377 | 0.366 | 0.364 | 0.343 | 0.38 |
| | Xinjiang | 0.472 | 0.477 | 0.499 | 0.502 | 0.472 | 0.452 | 0.424 | 0.422 | 0.423 | 0.392 | 0.454 |
| | **Mean** | **0.571** | **0.583** | **0.632** | **0.644** | **0.594** | **0.578** | **0.551** | **0.551** | **0.556** | **0.519** | **0.571** |
| | Liaoning | 0.565 | 0.578 | 0.615 | 0.639 | 0.618 | 0.605 | 0.59 | 0.597 | 0.592 | 0.55 | 0.595 |
| | Jilin | 0.51 | 0.511 | 0.541 | 0.539 | 0.535 | 0.55 | 0.54 | 0.539 | 0.552 | 0.526 | 0.534 |
| Northeast | Heilongjiang | 0.552 | 0.561 | 0.607 | 0.628 | 0.605 | 0.591 | 0.559 | 0.56 | 0.571 | 0.544 | 0.578 |
| | **Mean** | **0.542** | **0.55** | **0.588** | **0.602** | **0.586** | **0.582** | **0.563** | **0.565** | **0.572** | **0.54** | **0.542** |
| **Average EEs in Chinese provinces** | | **0.688** | **0.691** | **0.721** | **0.724** | **0.689** | **0.676** | **0.654** | **0.656** | **0.674** | **0.632** | **0.680** |

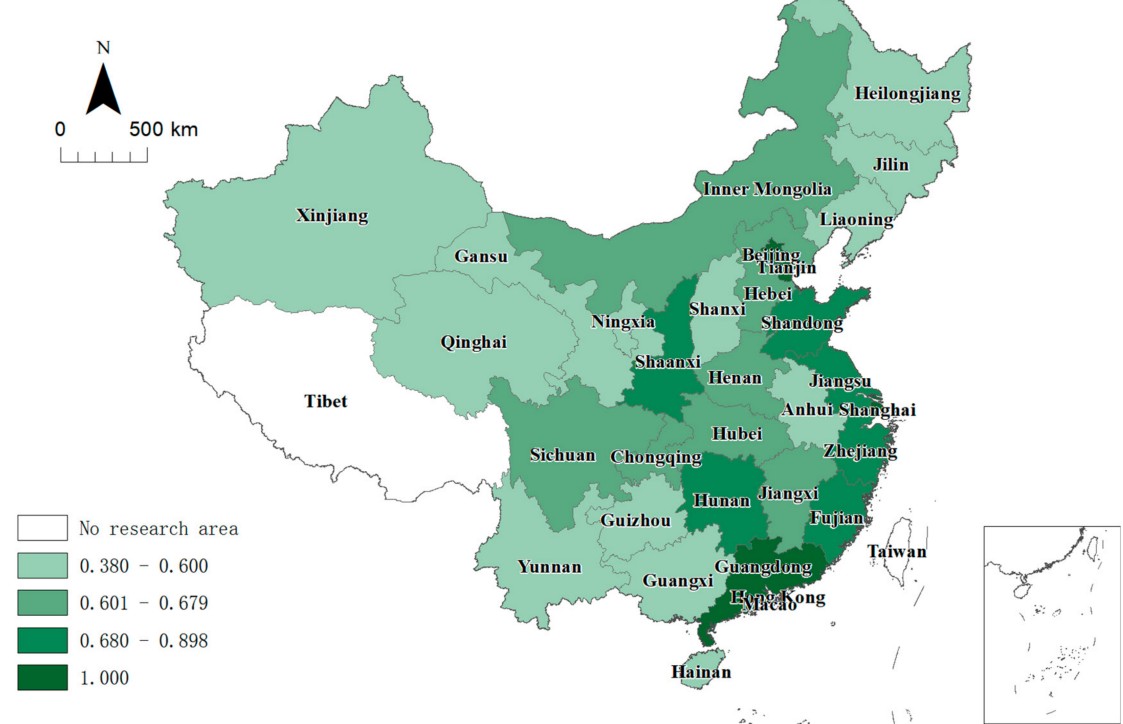

**Figure 3.** Average EE values in 30 Chinese provinces (2008–2017).

During the period considered, the overall national mean EE and those for the western and northeastern regions demonstrate inverted V-shaped trends with maxima realized in 2011. The eastern region shows a downward trend, while the eastern region presents an M-shaped fluctuation trend with two high points in 2011 and 2016 (Figure 4).

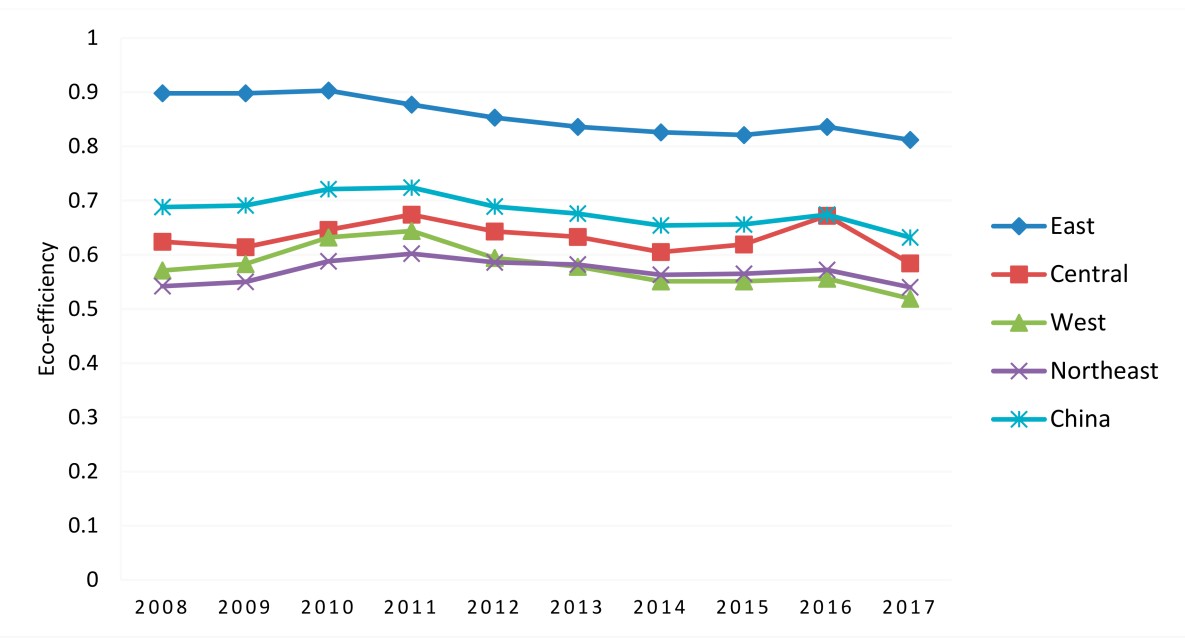

**Figure 4.** The evolutionary trend of EE in China and four regions from 2008 to 2017.

*5.2. Regional Characteristics of EE*

5.2.1. Eastern Area

Levels of EE are especially high in the eastern part of China. The efficiency values of Beijing, Tianjing, Shanghai, and Guangdong have always been 1. The eastern area has a better economic basis and enjoys geographical advantages. Since the economic reform, the Chinese government promotes an unbalanced development strategy in which the capital and policy considerations were focused on the Eastern area [89]. Meanwhile, new foreign production technologies and management methods were actively brought into the region [54], and these have had a positive effect in increasing technology stocks. With the exceptions of Hebei and Hainan, each province in the region has a high EE value far above the national average. Because of efforts to disperse non-capital functions, many secondary industries exhibiting high pollution and high energy consumption were transferred from Beijing to Hebei, leading to increases in production emissions in Hebei [23]. Hainan officials have failed to do the work of reducing emissions: emissions of $CO_2$, $SO_2$ and COD, and ammonia nitrogen in Hainan rose from 2008 to 2014, which inhibited improvement in the EE. Hainan showed a fluctuating trend in EE scores, while Hebei, Jiangsu, Zhejiang, Fujian, and Shandong exhibit decreasing trends during the study period.

5.2.2. Central Area

The mean EE value for the six provinces of central China is 0.624, which is close to the national average. The mean EE value in Hunan is higher than the national average. For Henan, it is the lowest, with an average value smaller than 0.6 during the study period, even though the EE exhibits an ascending trend until 2016. The cause of this phenomenon is that Henan did not succeed in reducing emissions until 2016. Shanxi also presents a downward trend. Shanxi has huge energy resources, and particularly possesses an abundance of coal; the traditional economic model brings severe environmental pollution and ecological damage to Shanxi [90,91]. The EE values of Anhui and Jiangxi were initially rising and then declined, with the maximum EE appearing in 2011. Though pollutant emissions in

Anhui and Jiangxi were largely under control after 2011, $CO_2$ emission remains high and exerts some restraint on the EE. The other two provinces in the central part of China show fluctuating trends during the study period.

### 5.2.3. Western Area

China's Western area has a comparatively low EE overall, with an average score of 0.571. The EE level of Shaanxi is above the national mean, while Ningxia and Xinjiang have the lowest EE levels in the country. This is the result of their unfavorable geographical situations and obsolete modes of production, technology, and management. The majority of the western provinces are important providers of resources and suffer from the inadequate accumulation of technology. Therefore, their economic development depends heavily on the consumption of resources and energy. In 2000, the State Council of China issued the "Western Development Strategy." Various preferential policies were implemented by the western provinces to attract industrial investment. Eastern provinces and foreign enterprises transferred many of their industries to the western area, and these are mainly labor and resource-intensive industries [92]. As a result, the economic growth pattern for the region has distinctive features of high investment growth, high consumption support, and high emission maintenance, and these exert negative impacts on regional EE.

### 5.2.4. Northeastern Area

The average EE value of the Northeastern area is 0.542, which is below the national average. The mean EE values of the region initially exhibit increasing trends and then suffer subsequent declines. In the observation period, the average EE values of Liaoning, Jilin, and Heilongjiang were 0.595, 0.534, and 0.578, respectively. The EE in Liaoning and Heilongjiang first rose and then declined, and the maximum EE appeared in 2011. Because of resource exhaustion, eco-environmental pollution, and the traditional economic structure, the Northeast area experienced a recession in the 1990s. As a result, in 2003 the Chinese central government implemented a strategy designed to revive the old Northeast industrial base and prescribed an array of preferential policies in the Northeastern area. However, the economic recovery of the region was still heavily dependent on fossil fuels and the release of pollutants [93].

### 5.3. Regression Results and Analysis

#### 5.3.1. Smulticollinearity Test

According to the correlation coefficient matrix between variables (Table 5), the correlation coefficients of some independent variables are greater than 0.7, which means that there may be significant multicollinearity among the variables concerned. Hence, a variance inflation factor (VIF) test is necessary. As shown in Table 6, all the values of VIF are less than 10, which indicates that there is no significant multicollinearity between the variables.

**Table 5.** The correlation test.

|  | In*EE* | In*DEL* | In*IS* | In*FTD* | In*TP* | In*UL* | In*PD* |
|---|---|---|---|---|---|---|---|
| In*EE* | 1 |  |  |  |  |  |  |
| In*DEL* | 0.5409 *** | 1 |  |  |  |  |  |
| In*IS* | 0.3723 *** | 0.5964 *** | 1 |  |  |  |  |
| In*FTD* | 0.7037 *** | 0.5694 *** | 0.5214 *** | 1 |  |  |  |
| In*TP* | 0.6892 *** | 0.6621 *** | 0.5329 *** | 0.6349 *** | 1 |  |  |
| In*UL* | 0.5991 *** | 0.9122 *** | 0.6726 *** | 0.7089 *** | 0.7090*** | 1 |  |
| In*PD* | 0.6723 *** | 0.4279 *** | 0.4240 *** | 0.6707 *** | 0.7078*** | 0.5293 *** | 1 |

Note. *** indicates significance at the 1% level.

#### 5.3.2. Spatial Autocorrelation Test

The study applied the Global Moran's I index to determine whether there was a spatial correlation in the EE values. The results are presented in Table 7, and the values of the index are positive and statistically significant at the 5% level during the observation year,

which means that the EE values of Chinese provinces exhibit a significantly positive spatial autocorrelation. In more concrete terms, provinces with high EE tend to be adjoining, while provinces with low EE adjoin other provinces with low EE.

**Table 6.** The variance inflation factor (VIF) test.

|  | **InDEL** | **InIS** | **InFTD** | **InTP** | **InUL** | **InPD** | **Mean VIF** |
|---|---|---|---|---|---|---|---|
| VIF | 6.63 | 1.86 | 2.80 | 3.01 | 9.66 | 2.51 | 4.41 |
| 1/VIF | 0.150881 | 0.537701 | 0.357327 | 0.331950 | 0.103474 | 0.398968 |  |

**Table 7.** Value of Moran's I of provincial EE in China (2008–2017).

| Year | Moran's I | Z-Score | *p*-Value |
|---|---|---|---|
| 2008 | 0.423 *** | 3.696 | 0.000 |
| 2009 | 0.417 *** | 3.653 | 0.000 |
| 2010 | 0.299 *** | 2.682 | 0.007 |
| 2011 | 0.257 ** | 2.364 | 0.018 |
| 2012 | 0.381 *** | 3.406 | 0.001 |
| 2013 | 0.343 *** | 3.102 | 0.002 |
| 2014 | 0.380 *** | 3.411 | 0.001 |
| 2015 | 0.318 *** | 2.890 | 0.004 |
| 2016 | 0.285 ** | 2.591 | 0.010 |
| 2017 | 0.327 *** | 2.977 | 0.003 |

*Note:* *** and ** indicate significance at the 1%, and 5% levels, respectively.

### 5.3.3. LM and Robust LM Tests

The tests described above confirm a notable autocorrelation phenomenon for EE in China. In such a case, the application of non-spatial panel data analysis methods might exhibit some biases [94]. With an abundance of caution, a series of statistical tests were carried out for deciding the appropriate model for empirically investigating the main determinants of EE.

The paper prepared the non-spatial panel models and performed the corresponding (robust) Lagrange multiplier (LM) lag and LM error tests. Table 8 presents the test results. In the LM tests, the null hypothesis of the non-spatially autocorrelated error term is rejected (statistics: 189.649, $p = 0.000$). The null hypothesis of the non-spatially lagged response variable is also rejected (statistics: 50.831, $p = 0.000$). In the robustness LM tests, the null hypothesis of the non-spatially autocorrelated error term is rejected (statistics: 139.848, $p = 0.000$), while the null hypothesis of the non-spatially lagged response variable is not rejected. Obviously, these results indicate that the spatial panel models are superior to non-spatial panel models. To ascertain the robustness of the model selection, the Wald and LR tests are applied subsequently.

**Table 8.** The results of Lagrange multiplier and Robust Lagrange multiplier test.

|  | **Spatial Error:** | **Spatial Lag** |
|---|---|---|
| Moran's I | 3.707 *** |  |
| Lagrange multiplier | 189.649 *** | 50.831 *** |
| Robust Lagrange multiplier | 139.848 *** | 1.030 |

*Note:* *** indicates significance at the 1% level.

### 5.3.4. Wald and LR Tests

The test results implied that the test statistics in Wald-lag, LR-lag, Wald-error, and LR-error are statistically significant (Table 9), which means that SDM was more appropriate

than SLM or SEM. The Hausman test indicates that fixed effects are not rejected. As shown in Equation (5), the explicit regression equation for SDM with fixed effects is:

$$
\begin{aligned}
InEE_{i,t} = {} & \rho WInEE_{i,t} + \beta_1 DEL_{i,t} + \beta_2 IS_{i,t} + \beta_3 FTP_{i,t} + \beta_4 TP_{i,t} + \\
& \beta_5 UL_{i,t} + \beta_6 PD_{i,t} + \theta_1 WDEL_{i,t} + \theta_2 WIS_{i,t} + \theta_3 W^*FTD_{i,t} + \theta_4 WTP_{i,t} + \\
& \theta_5 WUL_{i,t} + \theta_6 WPD_{i,t} + \varepsilon_{i,t} \; \varepsilon_{i,t} \sim N(0, \sigma^2_{i,t} \, In),
\end{aligned}
\tag{6}
$$

where $t$ stands for the year; $i$ denotes the province; EE represents the eco-efficiency; $\rho$ indicates the spatial autoregressive coefficient of dependent variables; $\beta$ is the unknown parameter; $\theta$ is the spatial autocorrelation vector of the explanatory variable; DEL, IS, FTP, TP, UL, EA, and PD express economic development level, industrial structure, foreign trade dependence, technological progress, urbanization level, and population density, respectively; $\varepsilon$ is the stochastic disturbance item.

**Table 9.** The regression results of Likelihood ratio test and Wald test.

| | Fixed Effects | Random Effects |
|---|---|---|
| Wald test spatial lag | 501.84 *** | 127.34 *** |
| LR test spatial lag | 134.89 *** | 134.89 *** |
| Wald test spatial error | 22.27 *** | 7.13 *** |
| LR test spatial error | 30.65 *** | 30.65 *** |

*Note:* *** indicates significance at the 1% levels.

The estimation results of the three fixed-effects models are displayed in Table 10. The LR test was applied to choose the most applicable model. The null hypothesis of the the spatial fixed effects are jointly insignificant is rejected at the 1% significance level. Moreover, The null hypothesis that the time fixed effects are jointly insignificant is rejected at the 1% significance level. Therefore, we choose SDM with spatial and time fixed-effects as the final empirical model for analyzing the factors affecting EE.

**Table 10.** The regression results of spatial Durbin model (SDM).

| | Spatial Fixed-Effects | Time Fixed-Effects | Spatial and Time Fixed-Effects |
|---|---|---|---|
| InDEL | 0.397 *** | 0.053 | 0.402 *** |
| InIS | 0.091 ** | −0.200 *** | 0.092 |
| InFTD | 0.026 * | 0.124 *** | 0.027 * |
| InTP | 0.080 * | 0.078 | 0.080 * |
| InUL | −0.105 | 0.125 | −0.127 |
| InPD | −0.046 *** | −0.040 * | −0.044 *** |
| WInDEL | −0.306 *** | −0.064 | −0.259 |
| WInIS | −0.008 | 0.141 | −0.003 *** |
| WInFTD | −0.024 | −0.095 *** | −0.041 |
| WInTP | −0.067 | −0.037 | −0.045 |
| WInUL | −0.046 | −0.035 | −0.012 |
| WInPD | 0.069 *** | 0.042 | 0.055 |
| Spatial rho | 0.772 *** | 0.684 *** | 0.701 *** |
| Variance sigma2_e | 0.005 | 0.010 | 0.005 |
| R-squared | 0.775 | 0.640 | 0.672 |
| Log-likelihood | 335.771 | 249.772 | 338.962 |

Note: ***, **, and * indicate significance at the 1%, 5%, and 10% levels, respectively.

### 5.3.5. Analysis Results

The economic development level played a significant role in promoting EE in China, which is consistent with the work of Zhu et al. [55], Dong et al. [58], and Liu et al. [80]. With improvements in economic development levels, local areas have more money to develop new technology and new products, introduce advanced technology and qualified personnel, upgrade product technology, and improve pollution disposal technology and management.

The correlation coefficient determined for industrial structure and EE was 0.092, but it did not pass the test of significance, which shows that the development of the tertiary industry had no significant effects on EE. With industrial transformation and upgrading in China, the correlation degree between industrial structure and environmental pollution

gradually weakened, which caused the effect of industrial structure on the emission reduction to be insignificant [95].

The impact of foreign trade dependence on EE is significant and positive. This illustrates that pursuing foreign trade can improve EE, which is consistent with our expectations as well as with the results of Xu [79], Liu et al. [80], and Zhou et al. [81]. In order to surpass the green barrier successfully and enter a broader international market, local Chinese enterprises continue to carry out multi-directional innovation in areas such as strategy, management, and technology, and this had a significant positive effect on EE.

Technological progress has a significant positive influence on EE at the 10% level, this is consistent with the work of Dong et al. [58], Zhou et al. [81], and Ahmad et al. [96], and with our expectations. However, the regression coefficient is not very high, which indicates that the promoting role of technological progress on EE in China is not high and has considerable room for improvement [97]. The proportion of R&D expenditures in GDP have risen from 1.54% to 2.13% during the study period [77], which leaves a gap with the 2.5% needed for an innovative country [98].

The regression coefficient for the level of urbanization is −0.127, but it is not evident. The gathering of production elements brought about by urbanization can trigger technological innovation [99], which can neutralize the adverse effects of pollution to some extent. At this stage of China's rapid urbanization, the national urbanization level has grown from 17.9% in 1978 to 58.5% in 2017, while the global urbanization level has also increased from 38.6% to 54.8% over the same period [7]. Therefore, China inevitably faces many ecological–environmental problems and should steadily promote urbanization to realize the fullest utilization of materials, energy, and information.

Population density has a significant negative impact on EE. This conclusion is consistent with the work of Zhu et al. [55], Diaz-Villavicencio et al. [87], and Ahmad et al. [92]. It shows that the population density has also brought more negative impacts on the ecological environment in China. A large number of people flowed into the eastern coastal areas after China issued the reform and opening policy, and efforts to protect the ecological environment in these areas should be further enhanced.

## 6. Conclusions and Policy Suggestions

The environmental harm caused by human activities constitutes a global challenge and requires a global response [100]. The article has used the EBM model with undesirable outputs, which can compensate for the weakness of radial and non-radial DEA methods, to evaluate Chinese EE and obtain more results than previous studies [16]; thus, we get a more accurate understanding of the overall level of the Chinese EE levels. Most provinces in China have a low level of EE; the EE in the east is the highest, followed by those in the middle and west, while that of the northeast is the lowest. These results agree with those of Dong et al. [54] and Liu et al. [76]. However, of greater concern, the paper finds that the mean EE values for 30 provinces in China exhibit decreasing trends since 2011, and $CO_2$ emissions continue to show an increasing trend during the period. This presents us with a warning: China is facing unusually severe environmental pollution, and the ongoing work of energy-saving and emission reduction must be continued, so there is a need to explore more effective measures for reducing emissions.

As shown above, establishing the key measures for conserving energy and reducing emissions is crucial. The article has explored the factors influencing EE in China using SDM regression, which showed that economic development level, foreign trade dependence, and scientific and technological progress all have noticeably positive effects on EE. Conversely, population density has significantly negative influences on EE. Based on these regression results, the paper offers some suggestions. As the high level of economic development provides a forceful guarantee for developing EE, the government should change the mode of economic growth and take the green development road. As for the policies of industrial structure, the Chinese government should optimize and upgrade industrial infrastructure. For instance, the government can guide the orderly transfer of industries, and forcefully

develop new and high-tech industries and modern service industry based on resource and environmental carrying capacity and regional functions. As for foreign trade, it is important to recognize that different foreign trade policies should be applied in different regions. With a high degree of dependence on foreign trade, the eastern region should increase the export of environmentally-friendly products and make them the primary export commodities. While vigorously developing international trade, the central, western, and northeast regions should pay attention to environmental protection, and increase the financial expenditure on industrial pollution control. Since technological progress can also help develop EE, governments and enterprises should implement a strategy of innovation-driven development and spend more on R&D. As for urbanization policy, it is important to accelerate reforms in urban land use and establish a rigorous farmland protection plan to increase urban land use efficiency. In the end, it is important to rationally control the regional scale of the population and optimize the structure of the population.

However, this study still contains some deficiencies that should be addressed in a later study. First, the paper focuses on the EE levels of administrative spaces, and future work should focus on the EE levels of specific industries or sectors such as agriculture, mining, manufacturing, construction, power generation, and transport. Moreover, the ultimate goal of EE research is to enhance the overall level of regional EE and achieve balanced and sustainable development among the economy, resources, and environment. Therefore, further research is needed on the convergence analysis of EE in China, and this will provide a theoretical basis for implementing regional balance and harmonious development.

**Funding:** This research was funded by the National Natural Science Foundation of China (No. 41925003); Ministry of Education Key Projects of Philosophy and Social Sciences Research (No. 18JZD029) and the UKRI's Global Challenge Research Fund (No. ES/P011055/1).

**Institutional Review Board Statement:** Not applicable.

**Informed Consent Statement:** Not applicable.

**Data Availability Statement:** Data was obtained from China official national statistical database.

**Conflicts of Interest:** The author declares no conflict of interest.

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
