# Peer review of "China’s Eco-Efficiency: Regional Differences and Influencing Factors Based on a Spatial Panel Data Approach"

_sustainability, doi:10.3390/su13063143_

Round 1
Reviewer 1 Report
see file attached

Author Response
Point 1: At this stage I do not recommend this paper for publication. It might be greatly improved before resubmission. The technical framework is not really new (and there are typos in the model), it should be better explained regarding some hypotheses. The regression part could be improved in focussing on othervariables.
The paper uses two relatively widely used techniques: the epsilon-based measure model (EBM) proposed in 2010 by Tone and Tsutui and the Spatial Durbin Model. In a rst stage the EBM is used to compute e?ciency indexes and then, in a second stage, indexes are regressed on a set of variables including foreign trade indicator, economic development level, population density, technological progress and industrial structure.
As re ected by the title of the paper, it is not clear what is the value added of this article.
The value added of the paper should be in the second stage regression, then either it is (for example) industrial structure of regions, foreign trade dependence or technical progress or a mix of these elements. Then, it should be made clear in the title for example "Does foreign trade in uence regional eco-effciency in China? A spatial approach.".
For example:
Li Yang, Ke-Liang Wang, Ji-Chao Geng, China's regional ecological energy effciency and energy saving and pollution abatement potentials: An empirical analysis using epsilon-based measure model, Journal of Cleaner Production, Volume 194, 2018, Pages 300-308
Use the same sample (up to 2015), about the same set of inputs and outputs and a DEA framework very similar to the one used in this paper. Another example is:
Zeng, L.; Lu, H.; Liu, Y.; Zhou, Y.; Hu, H. Analysis of Regional Di erences and In uencing Factors on Chinas Carbon Emission E?ciency in 20052015. Energies 2019, 12, 3081. https://doi.org/10.3390/en12163081
Response 1: We have changed the title“China’s Eco-efficiency: Regional Differences and Influencing Factors based on EBM model with undesirable outputs and spatial Durbin model” into the new tile“China’s Eco-efficiency: Regional Differences and Influencing Factors based on A spatial Pane Data Approach”. The new tile has the emphasis on spatial effect.
Point 2: This last paper (the author contributed to both papers) di ers from this one by the methodology used in the second stage: tobit model instead of spatial panel regression. More comments about the paper:
The introduction has to be revised given the choice of the variables in the second stage regression. China is a large economy in terms of GDP, the largest consumer of energy and also the largest producer of CO2 then China is an interesting case study. But, something that might be obvious for most people, but should be mentioned, is the reason why the regional dimension really matters. China is a very large and diverse country. China is almost as big as Europe! To follow the idea of the author a "one size ts all policy" is very likely to be less e?cient than a tailor made approach. The paragraph starting line 44 should be then rewritten. If there are regional policies implemented in China, they should be quoted, and, they certainly explain why they are di erent from one region to another (motivation). This justify why a spatial approach is interesting and might be relevant. There are regions that are more industrial than others, agriculture is more preponderant in other regions (this explain why GDP composition will be used in the second stage). Some regions trade more than others, etc. But there might be some common characteristics that explain partly common evolutions of e?ciency.
Response 2: The regional policies implemented in China has been quoted in the new manuscript. We have added the sentence“ In 2011, the Chinese government promulgated the main functional area planning, which has required to implement different environmental strategies to different regions.”in lines 54-56
Point 3: ALine 59 I do not understand what is mean by "objectiveness" and "precisions"? Do I have to understand that other methods provide imprecise results? That methods are subjective? I think that this kind of wording should be avoided. Page 67 it is wrong to say that the Tobit model is a non-panel model. See for example:
Bo E. Honor, Ekaterini Kyriazidou and J. L. Powell (2000) Estimation of tobit-type models with individual speci c e ects, Econometric Reviews, 19:3, 341-366, DOI: 10.1080/07474930008800476
Response 3: It is true that the line 59 has some problems. The sentence “Previous studies have shown that DEA has many advantages over SFA in terms of objectiveness and precision [12]”is inaccurate,which has been deleted.
The sentence“ the Tobit model is a non-panel model”is inaccurate,which has been revised as the new sentence“The traditional Tobit model ” in line 74
Point 4: Again from line 72, this paragraph should be changed the use of EBM is not a major contribution as it is relatively widely used in eco-e?ciency assessment but the second stage regression might be interesting. Line 78 remove the reference to precise numerical results.
Response 4: The sentence “An epsilon-based measure (EBM) model with undesirable outputs is utilized to evaluate the EE in China over the period 2008 to 2017, and this can compensate for the weakness of traditional DEA methods as well as deal with undesirable outputs [13-21, 25-27]” has been deleted.
The sentence “and achieves precise numerical results” has been deleted.
Point 5: Section 2 should include a table that could give an overview of quoted works: in the rst column name of authors, second column methodology, third columns inputs/outputs used,... It would be easier to get an overview of recent works and to see di erences with this paper.
Response 5: we have added the table 1 in line 127-128, and the Table 1 meets the above requirements.
Point 6: Section 3 methodology there are some problems. About the EBM there are typos in the model "x instead of " x , in the objective s b r should be s g r, in the set of constraints there is a ? in the third constraint that should not be there. But there are explanations missing. How do you compute the weights and the epsilons? What kind of a?nity indicator you use, do you use PCA? all these elements are very important and they partly drive the results. Then you have to elaborate a little bit more on that. In passing, it could be interesting to put a table with the values of epsilons as it indicates if you tend to a radial, directional 2 or slack based model. Some typos: line 161 and 162 subscripts for the epsilon, line 163 "bpo" line 166 "and and" the weights are missing. Also add that the weights are positive and they sum to 1 in the model. If the EBM is a plus then it could be compared with this previous paper (of the same author if I am correct):
Peng-jun Zhao, Liang-en Zeng, Hai-yan Lu, Yang Zhou, Hao-yu Hu, Xin-Yuan Wei, Green economic e?ciency and its in uencing factors in China from 2008 to 2017: Based on the super-SBM model with undesirable outputs and spatial Dubin model, Science of The Total Environment, Volume 741, 2020 At a rst glance the improvement is by considering radial and slack based approaches simultaneously. To my point of view there is no clear explanation or good examples to show to what extent it radically change the analysis. Regarding the spatial model, rst, you need to elaborate more on the Moran index, it is used to determine necessity of introducing the spatial econometric model (even if we have an a priori in favour of this model). You should put the formula. It might be of some interest to consider global and local Moran indexes. A possible extension could also be to compute di erent weight matrices as in:
Yang Kong, Weijun He, Liang Yuan, Zhaofang Zhang, Xin Gao, Yu'e Zhao, Dagmawi Mulugeta Degefu, Decoupling economic growth from water consumption in the Yangtze River Economic Belt, China, Ecological Indicators, Volume 123, 2021,
Response 6: Based on the research results of Chen et al. [1], and Wu et al. [2], and Ren et al. [3], we have revised the formula (1) in line 176-187
We have added the formula of Moran’ I in 190-200. Because the length of manuscript is not suitable for too long, we don't make local spatial autocorrelation analysis.
Point 7: At a rst glance the improvement is by considering radial and slack based approaches simultaneously. To my point of view there is no clear explanation or good examples to show to what extent it radically change the analysis. Regarding the spatial model, rst, you need to elaborate more on the Moran index, it is used to determine necessity of introducing the spatial econometric model (even if we have an a priori in favour of this model). You should put the formula. It might be of some interest to consider global and local Moran indexes. A possible extension could also be to compute di erent weight matrices as in:
Yang Kong, Weijun He, Liang Yuan, Zhaofang Zhang, Xin Gao, Yu'e Zhao, Dagmawi Mulugeta Degefu, Decoupling economic growth from water consumption in the Yangtze River Economic Belt, China, Ecological Indicators, Volume 123, 2021,
Response 7: We added the new content to show the advantages of EBM over the radial and slack based approaches in 161-176.
We added the formula of Moran’ I in 190-200. Because the length of manuscript is not suitable for too long, we don't make local spatial autocorrelation analysis.
Point 8: Section 4 data. The choice of inputs/outputs is similar to most studies, however I note that there are 11 outputs and inputs for 30 provinces. This number is relatively high and it is known that the more inputs/outputs used the less the discriminatory power of DEA (in general) is. It does not seem to be the case here. This could be mentioned. It is strange to put table 1 without statistics for others pollutant. The author could consider to have a table showing averages for all undesirable outputs per region rather than a single time series of CO2 emissions. Some indicators could be computed, for example, emissions per unit of GDP.
Response 8: We haved revised the Table 2, which can meet the requirements.
Point 9: ASection 4.2 factors in uencing EE. Here, again, a table might be usefull to have a quick look at similar studies and results obtained. In this part, I do not see in the choice of variable something really new but I think that there is room for improvement. Just a remak in passing, a table with correlation between inputs/outputs/factors would be interesting, it might be wise to see if there are not too much multicolinearity. But, my main idea is to test for other variables. To my knowledge you have by region the decomposition of the number of corporate rms into: State, collective, private and foreign holding (table 1-8 statistical yearbook of China). Do regions with more foreign rms are more or less e?cient? You also have an indicator of foreign investments. To capture economic/social development there is per capita income (table 6-17), consumption of durable goods (table 6-22). About technology there are data about patents (table 20-7 and following). What about development zones, are they equally distributed across the country? Is there an impact on effciency?
Averages are presented by regions, it would be interesting to see if rather than presenting averages, weighted averages (using GDP as weights) do not provide more "accurate" gures.,
Response 9: We have made the Smulticollinearity test in 400-412.
We have tried to add patents and the rate of foreign direct investment to GDP as the dependent variables, but the P-value of them are statistically non-significant. So they are not described in the paper.
Point 10: Section 4 data. The choice of inputs/outputs is similar to most studies, however I note that there are 11 outputs and inputs for 30 provinces. This number is relatively high and it is known that the more inputs/outputs used the less the discriminatory power of DEA (in general) is. It does not seem to be the case here. This could be mentioned. It is strange to put table 1 without statistics for others pollutant. The author could consider to have a table showing averages for all undesirable outputs per region rather than a single time series of CO2 emissions. Some indicators could be computed, for example, emissions per unit of GDP.
Response 10: We have revised the Table 2, which can meet the above requirements.
Point 11: I think that sections 5.1 and 5.2 should be merged, the idea is to present results by regions with tables showing individual data on efficiency by regions. Basically it means to split table 4 by regions and to provide totals by region.
Response 11: We have revised the Table 4, which can show individual data on efficiency by regions, which can meet the above requirements.
Point 12: Line 430 do we really need 7 figures after the dot? idem line 451.
Response 12: We have revised this, which are 3 figures after the dot in line 485 and line 505.
Point 13: The conclusion includes some statements that are not directly linked with the rest of the paper or at least is not in the model. For example, lines 486-490 what the author mean by "high-quality economic growth" and how it is measured in the model? Lines 494-498 the same remarks hold for environmentally product- What is the link with "pollution rst - renovation behind"? May be exploring data on general expenditure by regions might provides some insight: http://www.stats.gov.cn/tjsj/ndsj/2019/indexeh.htm table 7-6
Response 13: It is true that the lines 486-490 have some problems. The sentence “high-quality economic growth”is inaccurate,which has been deleted.
The sentence“ and change the external trade pattern of "pollution first, renovation behind”. has been revised as the new sentence“ and increase the financial expenditure on industrial pollution control. ” in line 553-554
References
Chen, C. F.; Sun, Y. W.; Lan, Q. Q.;Jiang, F. Impacts of industrial agglomeration on pollution and ecological efficiency-A spatial econometric analysis based on a big panel dataset of China’s 259 cities. Journal of Cleaner Production. 2020, 258, 120721. https://doi.org/10.1016/j.jclepro.2020.120721
Wu, P.; Wang, Y. Q.; Chiu, Y. H.; Li, Y.; Lin, T.Y. Production efficiency and geographical location of Chinese coal enterprises - undesirable EBM DEA. Resources Policy, 2019. 64, 101527. https://doi.org/10.1016/j.resourpol.2019.101527
Ren, Y. F.; Fang, C.L.; Li, G.D. Spatiotemporal characteristics and influential factors of eco-efficiency in Chinese prefecture-level cities: A spatial panel econometric analysis. J. Clean. Prod 2020. 260, 120787. https://doi.org/10.1016/j.jclepro.2020.120787

Reviewer 2 Report
The manuscript deals with the assessment of Eco-efficiency (EE) in China over the period 2008-2017.
The topic is interesting and has a well-defined focus on 30 Chinese provinces. The paper is structured in a coherent and logical way. The list of references seems adequate to me.
My comments are as follows.
- Please, add some more details about the source (or sources) of input data.
- 7, lines 224. Why the authors selected GDP as the desired output? Can you please motivate this choice?
Author Response
Point 1: please, add some more details about the source (or sources) of input data
Response 1: The sentence“The data on gross fixed capital formation, the total number of employees, and total water consumption, and urban construction land were taken from the China Statistical Yearbook (CSY)(2009-2018) [1]. The data on energy consumption come from the China Energy Statistical Yearbook (2009-2018) [2]. ”has provide data sources of input data.
Point 2: P7, lines 224. Why the authors selected GDP as the desired output? Can you please motivate this choice?
Response 1: As we can see from the Table 1 in line 127-128,many research has been applied GDP as the desired output. GDP measures the total financial value of all the goods and services produced in an economy over the course of one year. It's very suitable to select GDP as as the desired output

Reviewer 3 Report
Page 5, Line 163: why do you write 0 as o? It does not correspond to equation (1).
Page 5, lines 165-167: You need to rewrite this sentence. As such, it is not clear.
Page 5, lines 182-183: You need to rewrite this sentence. As such, it is not clear.
Page 6, line 191: It is "applies" instead of "applie".
Page 6, lines 198-199: There is a repetition with respect to your previous sentence (lines 195-196).
Page 6, line 211: It is "the article chooses" instead of "chose".
Page 7, Table 1: I do not understand the purpose of Table 1, with respect to what follows in the paper.
Page 9, lines 247-249: It is not the paper that selected but you as an author. You have to rewrite this sentence for more clarity. You say that you expect positive effects on EE. Where does this assumption come from?
Page 9, lines 252-253: Once again, it is not the paper that selected but you as an author.
Page 9, line 259: Again, it is not the paper that selected but you as an author.
Page 10, line 269: Again, it is not the paper that selected but you as an author.
Page 15, line 363: It is not the article that applied but you as an author.
Page 15, line 382: It is not the paper that prepared but you as an author.
Page 16, equation (4): You do not define the content of the equation.
Page 17, lines 414-416: You need to rewrite this sentence. As such, it is not clear.
Page 18, line 439: The sentence is not finished. You need to add something after "and".
Author Response
Point 1: Page 5, Line 163: why do you write 0 as o? It does not correspond to equation (1).
Response 1: It is 0, and we have changed “o”into“0”.
Point 2: Page 5, lines 165-167: You need to rewrite this sentence. As such, it is not clear. .
Response 2: It is true that the lines 165 to 167 have some problems. Based on Chen et al [1], Ren et al.[2], and Wu et al. [3],the sentence “α and β stand for the radial and non-radial programming parameters, which are combined by key parameters εx and εy” has been revised as the new sentence “εx stands for the set of radial α and non-radial slacks; εy denotes the set of radial β and non-radial slacks” in lines 185 - 191.
Point 3: APage 6, line 191: It is“applies" instead of "applie".
Response 3: We have changed “applie” into “applies”in line 231.
Point 4: lines 198-199: There is a repetition with respect to your previous sentence (lines 195-196).
Response 4: The sentence“W indicates spatial matrix if the provinces are adjacent, W = 1, otherwise, W = 0”does repeat my previous sentence,which has been deleted.
Point 5: line 211: It is "the article chooses" instead of “chose”.
Response 5: We have changed“the article chooses”into“the study chose”in line 249
Point 6: Table 1: I do not understand the purpose of Table 1, with respect to what follows in the paper.
Response 6: We have change the content of Table 1, which gives an overview of quoted works.
Point 7: lines 247-249: It is not the paper that selected but you as an author. You have to rewrite this sentence for more clarity. You say that you expect positive effects on EE. Where does this assumption come from?
Response 7: We have changed “paper” into “study”, and it is in a new position of line 283. Based on the previous sentence: The index of the industrial structure reflects the distribution of industry in an area. The higher the level of industrial structure, the more effective is the allocation of resources. Rational allocation of resources, to an extent, promotes stable and healthy development in the regional ecological environment. Therefore,we expect that industrial structure may have positive effects on EE.
Point 8: lines 252-253: Once again, it is not the paper that selected but you as an author.
Response 8: we have changed “article” into “study”, and it is in a new position of line 289.
Point 9: line 259: Again, it is not the paper that selected but you as an author.
Response 9: we have changed “article” into “study”, and it is in a new position of line 295.
Point 10: line 269: Again, it is not the paper that selected but you as an author.
Response 10: we have changed “article” into “study”, and it is in a new position of line 305.
Point 11: I line 363: It is not the article that applied but you as an author.
Response 11: we have changed “article” into “study”, and it is in a new position of line 414.
Point 12: line 382: It is not the paper that prepared but you as an author.
Response 12: It is true that the lines 382 have some problems.The sentence“ With an abundance of caution, we proceed to carry out a series of statistical tests with which we can decide the appropriate model for empirically investigating the main determinants of EE” is inaccurate,which has been revised as“With an abundance of caution, a series of statistical tests were carried out for deciding the appropriate model for empirically investigating the main determinants of EE. ”in 428-429.
Point 13: equation (4): You do not define the content of the equation.
Response 13: We has defined the content of the equation (6) in line 453-460.
Point 14:lines 414-416: You need to rewrite this sentence. As such, it is not clear.
Response 14: The sentence“The estimation results of the three SDMs with fixed-effects are displayed in Table 8. The LR test was applied to choose the most applicable SDM model. The null hypothesis of the time fixed-effects model is more appropriate than the spatial and time fixed-effects model is rejected (chi2(10) = 178.38, P = 0.000). Moreover, the test of the spatial and time fixed-effects model is more appropriate than the spatial fixed-effects model is not rejected (chi2(10) = -6.38, P = 1.000). ”is inaccurate,which has been revised as“The estimation results of the three fixed-effects models are displayed in Table 10. The LR test was applied to choose the most applicable model. The null hypothesis of the time fixed-effects model are jointly insignificant is rejected at the is rejected (chi2(10) = 178.38, P = 0.000). Moreover, the test of the time fixed-effects model are jointly insignificant is rejected (chi2(10) = -6.38, P = 1.000). ”
Point 15:line 439: The sentence is not finished. You need to add something after "and".
Response 15: The sentence “ This result is consistent with expectations and the conclusions of Zhu et al. [51] and Xing et al. [91]”is inaccurate,which has been deleted.
References
Chen, C. F.; Sun, Y. W.; Lan, Q. Q.;Jiang, F. Impacts of industrial agglomeration on pollution and ecological efficiency-A spatial econometric analysis based on a big panel dataset of China’s 259 cities. Journal of Cleaner Production. 2020, 258, 120721. https://doi.org/10.1016/j.jclepro.2020.120721
Wu, P.; Wang, Y. Q.; Chiu, Y. H.; Li, Y.; Lin, T.Y. Production efficiency and geographical location of Chinese coal enterprises - undesirable EBM DEA. Resources Policy, 2019. 64, 101527. https://doi.org/10.1016/j.resourpol.2019.101527
Ren, Y. F.; Fang, C.L.; Li, G.D. Spatiotemporal characteristics and influential factors of eco-efficiency in Chinese prefecture-level cities: A spatial panel econometric analysis. J. Clean. Prod 2020. 260, 120787. https://doi.org/10.1016/j.jclepro.2020.120787

Round 2
Reviewer 1 Report
Line 3 "Panel" and not "Pane".
Line 35 there is a "." missing at the end of the sentence.
Line 70 use "recent" instead of "new".
Line 118 "DMU" not "DUM".
Line 123 it is "NOx" not "CO2".
Line 127 in the table headings of columns 2 and 3 are inverted.
Line 127 last column fourth line "output" and not "outpu"
Line 177 in the model, last constraint there is lambda that should be removed before "=".
Line 185 the definition of epsilons is wrong then remove end of sentence line 185 up to ";" line 186. Keep end of line 186 and line 187. [As previously mentioned it would have been nice to explain how the epsilons are computed...]
Line 191, formula, "Global" not "Glbal"
Line 314, table 4 column 2 there is a line "Mean" that should be "East" and mean for each region should be in bold.
(...)
Reviewer 3 Report
Thank you for the revised version of the paper.